# Diversity and Reassortment Rate of Influenza A Viruses in Wild Ducks and Gulls

**DOI:** 10.3390/v13061010

**Published:** 2021-05-27

**Authors:** Yulia Postnikova, Anastasia Treshchalina, Elizaveta Boravleva, Alexandra Gambaryan, Aydar Ishmukhametov, Mikhail Matrosovich, Ron A. M. Fouchier, Galina Sadykova, Alexey Prilipov, Natalia Lomakina

**Affiliations:** 1Chumakov Federal Scientific Center for the Research and Development of Immune-and-Biological Products, Village of Institute of Poliomyelitis, Settlement “Moskovskiy”, 108819 Moscow, Russia; postni.yulya@ya.ru (Y.P.); narmoriel5991@gmail.com (A.T.); e@boravlev.mccme.ru (E.B.); sue_polio@chumakovs.su (A.I.); 2Institute of Virology, Philipps University, Hans-Meerwein-Str. 2, D-35043 Marburg, Germany; m.matrosovich@gmail.com; 3Department of Virology, Erasmus Medical Centre, Dr. Molewaterplein 50, 3015GE Rotterdam, The Netherlands; r.fouchier@erasmusmc.nl; 4The Gamaleya National Center of Epidemiology and Microbiology of the Russian Ministry of Health, 123098 Moscow, Russia; gksadykova@gmail.com (G.S.); a_prilipov@mail.ru (A.P.); nflomakina@gmail.com (N.L.)

**Keywords:** avian influenza, reassortment, diversity

## Abstract

Influenza A viruses (IAVs) evolve via point mutations and reassortment of viral gene segments. The patterns of reassortment in different host species differ considerably. We investigated the genetic diversity of IAVs in wild ducks and compared it with the viral diversity in gulls. The complete genomes of 38 IAVs of H1N1, H1N2, H3N1, H3N2, H3N6, H3N8, H4N6, H5N3, H6N2, H11N6, and H11N9 subtypes isolated from wild mallard ducks and gulls resting in a city pond in Moscow, Russia were sequenced. The analysis of phylogenetic trees showed that stable viral genotypes do not persist from year to year in ducks owing to frequent gene reassortment. For comparison, similar analyses were carried out using sequences of IAVs isolated in the same period from ducks and gulls in The Netherlands. Our results revealed a significant difference in diversity and rates of reassortment of IAVs in ducks and gulls.

## 1. Introduction

Wild aquatic birds of the orders Anseriformes (ducks, geese, and swans) and Charadriiformes (gulls, terns, waders, and plovers) are natural hosts of IAVs [1]. In these birds, 16 antigenic subtypes of hemagglutinin (HA) and 9 antigenic subtypes of neuraminidase (NA) in various combinations were found [2]. IAVs of H1–H12, H14, and H15 HA subtypes mainly circulate in ducks; these IAVs replicate in the intestine, cause asymptomatic infections, and transmit by the fecal-oral route through the water [1]. IAVs of H13 and H16 HA subtypes mainly circulate in gulls, causing epidemics among chicks in densely populated breeding colonies. These gull IAVs replicate in the intestine and the respiratory tract, and can transmit both by the fecal-oral route and through the air [3,4,5]. Thus, the patterns of replication and transmission differ between IAVs of ducks and gulls.

IAVs in domestic poultry evolve from IAVs of wild aquatic birds. IAVs of poultry can occasionally transmit and initiate host-specific evolutionary lineages in pigs, humans, and other mammals [1,2]. Long-term adaptation in ducks enables efficient replication and release of the virus into the environment without significant pathogenic effects to the host [2]. In gulls, the virus also replicates asymptomatically, while in poultry it may evolve towards increased pathogenicity [3]. In mammals, the influenza virus usually causes non-fatal disease [1].

The evolution of IAVs is caused by mutations and reassortment of viral genes. Reassortment occurs when a single host cell is infected with different IAVs [6]. Reassortment plays a central role in the microevolution of influenza viruses. In wild ducks, there is a continuous mixing of all eight genome segments due to reassortment. This process may be less intense in secondary hosts.

Reassortment between IAVs of different hosts, including human viruses, contributes to initiation of influenza pandemics [7]. However, once emerged, human IAVs of a novel lineage evolve as a whole, without reassortment. Only intra-subtype reassortants between co-circulating clades are viable and persist in the population on a small spatial-temporal scale [8,9,10,11,12]. Rarely, reassortants between H1N1 and H3N2 were found. For example, prior to the 2009 pandemic, clinical IAV isolates of the H1N2 subtype were described [13,14,15,16]. However, all these cases are rather the exception to the rule. Such reassortants are not sufficiently viable [17]. Studies of H1N2 reassortants in the human population showed that they are under negative selection [18,19]. Reassortment of IAVs in pigs is an example of the formation of new variants. Pigs were called the “mixing vessel” of IAVs because they can be infected with both avian and human viruses and enable their reassortment [20]. The IAVs that had been formed in pigs initiated the 2009 pandemic and, possibly, other pandemics. In recent decades, numerous new lineages of reassortant viruses were discovered in pigs [21,22,23,24]. However, IAVs of the same evolutionary lineage can circulate in pigs for many years. The “classical” lineage of swine IAVs, originating from the precursor of the 1918 “Spanish” influenza, remained genetically stable until the 1980s [25,26]. In horses, two lineages of influenza viruses (H7N7 and H3N8) evolved for decades [27].

Several lineages of highly pathogenic and low pathogenic IAVs circulate in chickens. Asian H5N1 and H7N9 IAVs are reassortants. They all carry a cassette of internal genes derived from chicken H9N2 viruses [28,29]. However, after initial formation, such variants circulate largely without reassortment. Chicken viruses of the H5 and H9 subtypes circulate in Egypt together, and co-infection with these viruses was often observed. Nevertheless, reassortment of these subtypes has not been reported yet [30]. Introduction of the H5N1 IAVs into ducks and their reassortment with duck viruses of other subtypes led to the emergence of novel H5N3, H5N6, and H5N8 IAV lineages [31,32,33].

The reassortment of IAVs in their natural animal hosts has a different pattern than in all secondary host species. It is impossible to detect persistent evolutionary lineages of duck IAVs containing stable constellations of all eight gene segments, as duck IAVs constantly exchange segments during mixed infections with neither significant fitness costs nor selective advantages of newly emerging combinations of segments [34].

Gulls are usually considered natural hosts of influenza viruses along with ducks. IAVs of almost all subtypes were found in gulls. However, the H1-H12 and H14 subtypes were only rarely isolated from gulls, and, as a rule, the gull viruses were evolutionarily close to the contemporary duck IAVs that circulated in the same region. These cases can be classified as occasional spillovers of duck H1-H12 and H14 IAVs to gulls. By contrast, the IAVs of H13 and H16 subtypes were almost exclusively detected in gulls and terns [35,36,37].

Although gene reassortment has been found among gull viruses, a relatively stable gene cassette containing PB2, PA, M, NS, NP, and PB1 gene segments has been formed in gulls and could persist for at least four years [38,39]. All gene segments of gull H13 and H16 IAVs are phylogenetically distant from segments of duck IAVs [35,36,40].

Recent studies have shown that reassortment can occur in any living system (cell culture, chicken embryos, laboratory animals) if the dose of infection is large enough to provide simultaneous access of different viral particles to the host cell [41,42,43]. However, the emergence of new variants of IAVs via reassortment in some hosts, such as humans, occurs once in about a decade, whereas in other hosts, such as ducks, the reassortment occurs continuously. This notion raises the key question of whether or not the rate of reassortment of IAVs differs significantly between viral host species [6].

The aim of our work was to compare the IAV reassortment rate in wild ducks and gulls. From 2006 to 2019, we isolated IAVs from wild birds resting in a city pond in Moscow, Russia during their autumn migration. Thirty-seven duck IAVs and one duck-origin spillover H6N2 gull isolate were fully sequenced. We did not find stable IAV genotypes perpetuated from year to year. For comparison, a similar study on genome constellations was carried using sequences of duck and gull IAVs isolated in the same years in The Netherlands.

## 2. Materials and Methods

### 2.1. Viruses

Fresh feces of mallard ducks and gulls were collected in 2006–2019 on the shore of a city pond in Moscow. Feces were suspended in 1 mL of phosphate-buffered saline (PBS) supplemented with 0.4 mg/mL gentamicin, 0.1 mg/mL kanamycin, 0.01 mg/mL nystatin, and 2% MycoKill AB solution (PAA Laboratories GmbH, Pasching, Austria). The suspension was centrifuged for 10 min at 4000 rpm, and 0.2 mL of the supernatant were inoculated into 10-day-old chicken embryos. Allantoic fluid was collected after 48 h and tested by hemagglutination assay with chicken red blood cells. Positive samples were taken for further passaging. All isolated IAV strains are stored in the virus repository of the Chumakov Federal Scientific Center (Moscow, Russia). Full names, designations of the viruses, and GenBank accession numbers are given in Appendix A.

### 2.2. Sequencing

Viral RNA was isolated from the allantoic fluid using QIAamp Viral RNA mini kit (Qiagen, # 52904) according to the instructions of the manufacturer. Reverse transcription was carried out at 42 °C for 1 h in a 25 μL reaction mixture containing 8 μL RNA, 1 μL uni12 primer with a concentration of 50 ng/μL (13.5 pmol/μL), 10 μL water, 1 μL 10 mM dNTP, 5 µL 5× buffer and 100 units of MMLV (Alpha-Ferment Ltd., Moscow, Russia). The resulting cDNA was used in PCR with specific terminal primers to synthesize full-length genome segments. The amplified fragments were separated by electrophoresis in 1–1.3% agarose gel in the presence of ethidium bromide and were eluted from the gel with a Diatom DNA Elution kit (Isogen Laboratory Ltd., Russia, Moscow # D1031). Sequencing reactions were performed with terminal or internal primers [44] using the BrightDye™ Terminator Cycle Sequencing Kit v3.1 (Nimagen, The Netherlands), followed by analysis on an ABI PRISM 3100-Avant Genetic analyzer (Applied Biosystems 3100-Avant Genetic Analyzer, Foster City, CA, USA,). For assembly and analysis of nucleotide sequences, the Lasergene software package (DNASTAR, Inc. Madison, WI, USA) was used.

### 2.3. Downloading of Sequences and Evolutionary Tree Construction

The complete nucleotide sequences of IAV internal genes (PA, PB1, PB2, NP, MP, and NS) and external genes (H1, H3, H4, H5, N1, H6, H11, N1, N2, N3, N6, and N8) were downloaded from the Influenza Research Database (https://www.fludb.org, accessed on 25 April 2021). The selected sequences were aligned by the MUSCLE method using the software package MEGA X (https://www.megasoftware.net/, accessed on 25 April 2021) [45]. Time-scaled trees were generated for each internal segment with known isolation dates using BEAST under GTR model with 1000 bootstrap replicates [46]. A strict molecular clock model was chosen for all segments.

### 2.4. Classification of Gene Variants and Detection of Gene Reassortants

All lineages and subordinate lineages were classified according to the topology of the phylogenetic trees using the approach described in [47], but with more detail. The major clades on each gene tree were defined by strong bootstrap support (>95%) and numbered. The minor clades within the major clade were designated by corresponding number and a letter.

## 3. Results

During the autumn periods of 2006–2019, feces of wild waterfowl were collected on the bank of a pond in Moscow city, and IAVs were isolated. Over 14 years, about 4000 samples were collected, and 38 strains of influenza A viruses of subtypes H1N1, H1N2, H3N1, H3N2, H3N6, H3N8, H4N6, H5N3, H6N2, H11N6, and H11N9 were isolated and completely sequenced (Table 1 and Appendix A). All isolates replicated in chick embryos to high titers, were infectious and immunogenic in mice, although these animals were generally not killed by the viruses. Five viruses tested were non-pathogenic in chickens [48,49].

The subtypes of isolates differed in different years. Until 2013, IAVs with hemagglutinins H3 and H4 dominated. These subtypes were no longer isolated after 2014 and were substituted by IAVs of the subtypes H1N2, H1N1, and H11N6. This change in the virus isolation pattern correlated with data from databases on the isolation of IAVs from ducks in Europe.

### 3.1. Evolutionary Relationships of Gene Segments

To study evolutionary relationships of gene segments, we expanded the set of IAVs isolated in Moscow by including the 191 mallard IAVs isolated in the Netherlands in the same period of time (2006–2019) [39]. We built phylogenetic trees for internal gene segments; clades/subclades on each tree were identified and numbered (Appendix A).

The A/gull/Moscow/3100/2006 (H6N2) was not fundamentally separated from mallard IAVs. Thus, for all gene segments of the gull isolate, closely related variants can be found among the duck viruses (Table 1 and Appendix A). A majority of IAVs isolated from mallards in Moscow had different gene constellations. The gene segments of the same clade/subclade were perpetuated in duck IAVs over several years. For example, PB2 of clade 11 was detected in 2009, 2010, 2013, 2014, 2018, and 2019 (Table 1).

Importantly, although segments of the same clade/subclade were found in duck IAVs isolated in different years, these viruses never preserved their full genome constellations and always contained a unique mixture of segments that belonged to different clades/subclades. Although four pairs of IAVs with identical constellations were detected among the Moscow viruses (Table 1), the viruses of each pair were isolated almost simultaneously and in the same place. These results support previous conclusion of Dugan and colleagues [34] that IAVs circulate in wild birds as a pool of gene segments, which reassort extensively and appear in a new combination each year.

### 3.2. Diversity of Gene Segment Constellations of IAVs in Ducks and Gulls

To compare the rate of natural reassortment of IAVs in ducks and gulls, we analyzed viruses isolated from ducks and viruses isolated from gulls in the Netherlands in the same period of time (2006–2019) [39]. The gull viruses were almost exclusively isolated from black headed gulls and were represented by 252 viruses of the H13 subtype, 94 viruses of the H16 subtype, and 11 mixed isolates.

The evolutionary trees for the gene segments were generated, and the clades and subclades were numbered (Appendix A). The data on genome constellations of 228 duck IAVs viruses and 357 gull IAVs are presented in Appendix A; selected data are shown in Figure 1 and Table 3. Among 228 duck IAVs, we identified 187 distinctive genotypes. Pairs of isolates that matched in all gene segments, as a rule, were isolated on the same day.

In only three cases, pairs of IAVs listed below with matching genomes were isolated in different years. They were: d/N/20/2012|H1N1 and d/N/5/2013|H1N1, d/N/48/2011|H7N1 and d/N/31/2012|H7N1, d/N/71/2008|H10N7 and d/N/1/2009|H10N7.

Table 2 shows the diversity of IAVs of different subtypes, calculated as the ratio of the number of mismatched variants of genomes to the number of IAV strains in the group. The diversity is very high among the H1, H2, H3, H4, H5, H6, and H11 subtype IAVs, with 83 to 100% of the virus genomes representing unique genome constellations (Appendix A). In contrast, gull IAVs were represented by 84 genome constellations per 252 H13 isolates (33% of genomes unique) and 38 constellations per 94 H16N3 isolates (40% of genome unique) (Appendix A).

Evolutionary histories of the internal gene segments of H13 and H16 IAVs are shown in Figure 2 with H16 IAVs colored in red. Non-persistent reassortment events are clearly visible on these trees as single strains of one color surrounded by strains with different color. Transition of gene segment from one subtype to another is also clearly visible as the alternation of groups of different colors on the same evolutionary branch. However, it is difficult to study the constellation of all gene segments on the trees, as this requires simultaneous comparison of patterns on different trees. To facilitate the analysis, we compiled tables in which we designated the clade numbers of the gene segments of all viruses in the same way as it was done for duck IAVs.

The gull virus isolates are naturally separated into large groups of closely related viruses, matching by subclades of all genes. Apparently, these groups represent variants of the same virus that caused the epidemic, e.g., in the nesting colonies. The genomic constellations of such variants are not very stable, they can vary from group to group and can change from year to year. Nevertheless, in some cases, clusters of the same variants of gene segment persist in the subsequent seasons. For example: HA, NA, PB2, PA, and NP of the H13N2/2016 isolates are direct descendants of the H13N2/2014 isolates; PB1, MP, and NS of both virus groups are descendants of the common close ancestors, HA, PB2, PB1, PA, and NP H13N2/2009 are descendants of H13N8/2008 (Appendix A).

The internal gene segments of H16N3 viruses usually do not match the corresponding segments of H13 viruses; however, this rule is not very strict. Some of the exceptions are listed below.

H16N3/2008, H13N2/2009 and H13N2/2010 shared PA.

In 2007, H13N6 and H16N3 had matching PA.

H13N6/2007 and H16N3/2008 shared NP.

In 2011, H13N6 and H16N3 had matching NP (complete data are given in Appendix A).

The trees are based on all published full-length sequences of H13 and H16 gull viruses isolated in The Netherlands. The route is placed on A/Black-headed gull/Netherlands/1/2000 (H13N8). Red color depicts sequences of H16 IAVs. Scale bars, 0.02. The trees were made using FigTree v1.4.4 (http://tree.bio.ed.ac.uk/software/figtree/, accessed on 25 April 2021).

In some seasons, large groups of gull IAVs had highly homologous gene segments, suggesting that all these IAVs represented descendants of the same precursor circulated during this season. The presence of such groups in our sets of gull IAVs allowed reliable identification of reassortant viruses, which clearly differed from the other isolates in the group by the origin of either one or a few gene segments (see Appendix A; some of data are given in Table 3).

For example, one isolate from the group of H13N6/2014 IAVs, bhg/16/2014, contained PB2 from H16N3/2014. Bhg/31/2009/H13N2 is a reassortant, in which all internal genes coincide with the H13N6 viruses of the same year, and only NA belongs to another clade (Appendix A).

Of particular interest are two H13N3 IAVs, bhg/17/2009 and bhg/8/2011, which contained heterologous N3 neuraminidase from H16N3. The former virus shared PB2, PA, NP, and NA with H16N3/2009, the remaining 4 segments were shared with other H13N6/2009 IAVs. Seven gene segments of bhg/8/2011 matched corresponding segments of H16N3/2011, only the HA being shared with H13N8/2011(Table 3 and Appendix A).

One can conclude that two H13N3 isolates originated from H16N3 IAVs, in which HA and a number of other genes were replaced by gene segments of H13 IAVs via reassortment. Apparently, such reassortants are not viable enough; they were found in only two isolates despite the fact that there were seven mixed isolates in the same set (Appendix A).

Some of the H16N3 IAVs represent reassortants containing internal gene segments of the H13 viruses. Thus, among 94 H16N3 viruses analyzed, we recognized 17 non-persistent reassortants, with 13 of them carrying gene segments of H13 viruses (Appendix A). Among 252 H13 viruses studied, we found 14 non-persistent reassortants, nine of which acquired at least one gene segment from H16 viruses. About 10% of the gull viruses isolated within one season were reassortants, this number being of the same order as the number of mixed isolates (3%) (see Appendix A).

Thus, although partial reassortment of gene segments can be observed over years among H13 and H16 IAVs, constellations of several segments remained stable for prolonged periods of time. The observed pattern is very different from what we observed in duck viruses of H1-H6 and H11 subtypes. No stable gene constellations could be detected in the sets of duck IAVs studied.

## 4. Discussion

Comparison of duck and gull viruses showed a large difference in the detection rate of reassortment in the main duck subtypes (H1, H2, H3, H4, H5, H6, and H11) and the gull subtypes H13 and H16. In gull viruses, reassortment was a fairly frequent, well-recorded phenomenon, while in duck viruses, gene mixing was so intense that it was almost impossible to find viruses with the same genome constellation. There may be several explanations for this difference.

A key difference between gulls and ducks is that gulls breed in dense colonies with much mixing of chicks, whereas ducks are dispersed during breeding. As virus spreads among the gull chicks in nesting colonies, a single newly introduced variant can infect many birds. Large numbers of nearly identical strains can be isolated at this time. In the separated gull colonies, an outbreak of another variant of the virus may occur. Thus, three clusters of closely related isolates from 2008 (one H16N3 cluster and two H13N8 clusters) represent independent epidemics in three different colonies [5]. Sometimes, the virus is carried from one colony to another, leading to emergence of mix-isolates and reassortants. However, these events are rather exceptions than the rule. There are even lower chances of mixed infections in gulls during their seasonal migration, as gulls do not form large flocks during migration.

Because ducks do not breed in colonies, transmission among young ducks is limited during breeding. On the other hand, during the moult and the fall flight, ducks from multiple breeding areas may mix, creating ideal conditions for mixed infections with distinctive influenza viruses. On the pond where the Moscow duck viruses were isolated, hundreds of mallards accumulate in the fall gathering along the edges of the pond, where children throw pieces of bread. Ducks arrive from the north of Europe [50,51]. Mallards spend about 2 months on this pond. The first birds arrive in mid-September, followed by constantly increasing numbers of birds. As shown by Wille and colleagues, ducks can be infected sequentially by several variants of IAVs, and excrete virus intensively in their feces. Therefore, each introduced virus multiplies and infects other ducks, thus promoting multiple infections and reassortments [52].

The second factor affecting the rate of reassortment is the pressure of natural selection, which sweeps unsuccessful combinations. In the secondary hosts, such as chickens and humans, this factor is probably the main reason for the persistence of certain optimal gene constellations over the years. For example, during the epidemics in humans, when two subtypes co-circulate, co-infection and even reassortment is quite possible [13,14,15,16], but reassortants are usually less viable than the parental variants and do not become fixed in the population [18,19].

Probably, the same reason explains the stability of the genomes of highly pathogenic influenza viruses. After moving from wild birds to poultry, the viruses adapt to a new host and a new route of virus transmission. The adaptation of IAVs to chickens is associated with an increase of the virus pathogenicity [53]. This effect can be explained, at least in part, by the IAV evolution towards efficient transmission in infected poultry owing to cannibalism (that is, pecking and eating of sick individuals). A characteristic feature of chicken influenza is the selection of IAVs with polybasic cleavage site in the HA, which enables the virus to infect endothelial cells and to cause generalized infection [54,55]. Acquisition of new properties requires the coordinated evolution of all genes. In each of the evolutionary branches of IAVs that adapt to chickens, unique concerted changes in the genes could take place, so that exchange of some segments by reassortment can destroy an interplay between the genes and/or their products and make the virus less viable. Naturally, such reassortants will be swept by natural selection, leading to the persistence of specific constellations of the gene segments.

In duck viruses, stable genome constellations are largely absent [34]. Gull viruses, by contrast, have semi-stable genome constellations, among them, stable combinations of HA and NA. The H16 HA is tightly associated with the N3 NA. The NAs of IAVs with H13 HA (N2, N6, and N8 NAs) separated in the course of evolution from the ancestral viruses of ducks and adapted to the viruses of gulls. The variants with H13 and N3 did not form stable evolutionary lineages, probably, they are not viable enough. The functional balance of HA and NA is an essential element of the viability of IAVs [56]. The receptor specificity of gull IAVs differs from that of duck IAVs. In contrast to duck IAVs, gull IAVs efficiently bind to fucosylated receptors. Unlike all other IAVs of aquatic birds, H16 IAVs recognize both 2–3 and 2–6 sialyl-galactose receptors, being in this respect more similar to swine viruses than to duck viruses [57]. Likely, the neuraminidase of H16N3 viruses is specifically adapted to H16 HA.

The internal gene segment of the H13 and H16N3 viruses, as well as the HA and NA segments, represent separate evolutionary branches adapted to gulls. The internal genes of the H13 and H16 viruses are still interchangeable, but tend to form relatively stable constellations in each subtype.

## 5. Conclusions

Duck AIVs represent a unique variant of the symbiosis of the virus with the host, where the virus does not persist as a specific genome, but as a pool of genes, from which new genomic combinations are constantly formed [34]. However, in non-duck species, including gulls, a different evolution scenario is common, when the virus evolves in the host in the form of a whole genome.

## Figures and Tables

**Figure 1 viruses-13-01010-f001:**
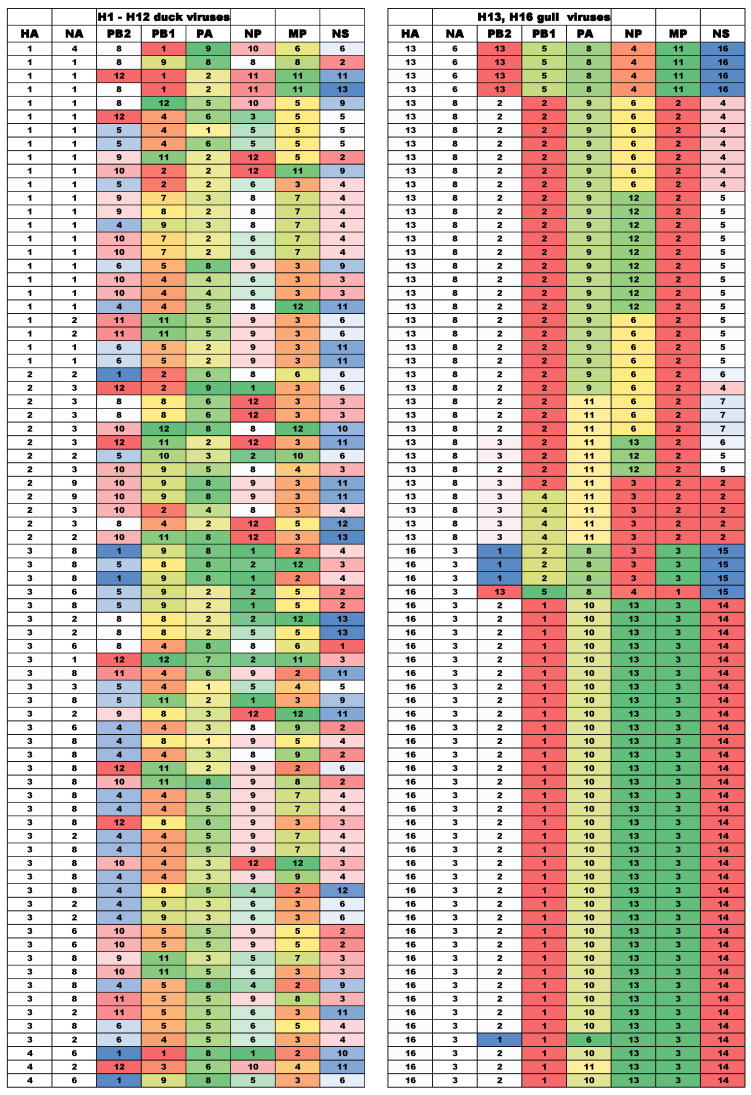
Genome constellations of IAVs isolated from ducks and gulls in The Netherlands in 2007–2009 (colors in each column depict clades of the indicated gene segment on the phylogenetic tree (see Appendix A). The names of the strains are not shown in the figure for clarity, they are shown in the Appendix A).

**Figure 2 viruses-13-01010-f002:**
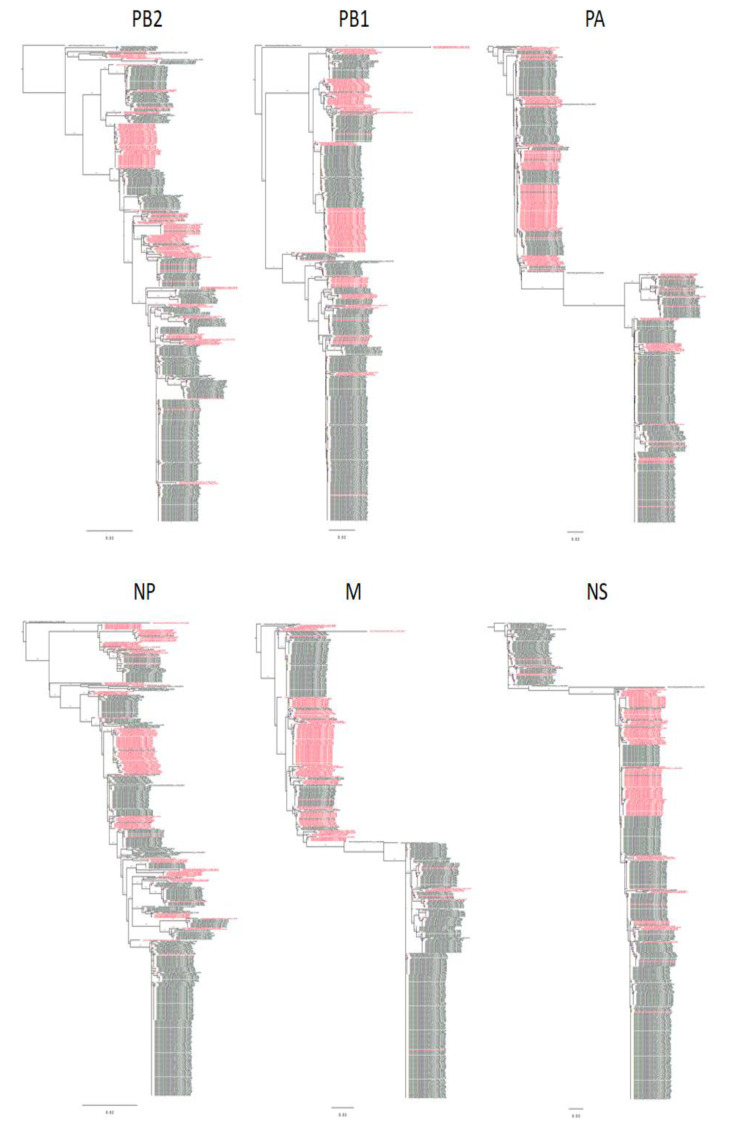
Evolutionary history of six internal gene segments of H13 and H16 AIVs.

**Table 1 viruses-13-01010-t001:** Clades and subclades of the gene segments of IAVs isolated in Moscow.

Isolation Date	Strain Name ^a^	Subtype	Clade/Subclade ^b^					
			PB2	PB1	PA	NP	M	NS
15 October 2019	M/5743/2019	H1N1	6	5	2a	10b	4E	B11
15 October 2019	M/5744/2019	H1N1	6	5	2a	10b	4E	B11
1 October 2013	M/4970/2013	H1N1	6	5	8a	10c	4D	B9d
17 October 2018	M/5586/2018	H1N2	11	12A	5a	10b	4E	6a
1 November 2018	M/5662/2018	H1N2	11	12A	5a	10b	4E	6a
13 October 2008	M/4203/2010	H3N8	4	4C	3a	9B	10	2b
26 November 2010	M/4238/2010	H3N6	4	4C	3a	9B	10	2b
13 October 2010	M/4298/2010	H3N8	4	9C	1	10e	5e	4d
27 September 2011	M/4494/2011	H3N8	4	4A	5b	10d	7	4c
10 October 2011	M/4681/2011	H3N8	4	4A	5b	10d	7	4c
1 October 2011	M/4524/2011mix	H3N2	4	4C	5b	10d	7	4c
1 October 2011	M/4524/2011mix	H3N8	4	4C	5b	10d	7	4c
1 October 2012	M/4780/2012	H3N8	4	9C	5a	4	3	B12
1 October 2012	M/4788/2012	H3N8	4	4C	3a	10d	10	4c
4 September 2009	M/3806/2009	H3N8	11	4C	6b	10d	3	B11
1 October 2014	M/5037/2014	H3N8	11	5	5b	10a	9	3c
13 October 2008	M/3556/2008	H3N1	12	13	7	2B	12	3d
13 November 2010	M/4242/2010	H3N8	12	12A	2b	10d	3	6a
10 November 2011	M/4661/2011	H3N8	12	9C	6b	10d	4E	3e
19 September 2009	M/3740/2009	H4N6	2	9C	1	10e	5e	B11
4 September 2009	M/3799/2009	H4N6	2	9C	1	10e	1	B11
19 September 2009	M/3735/2009	H4N6	3	9C	1	10d	1	B11
31 October 2012	M/4781/2012	H4N6	4	4C	3a	10d	7	3e
17 October 2012	M/4843/2012	H4N6	4	4C	3a	3	7	4c
10 October 2012	M/4771/2012	H4N6	6	4A	3a	3	7	4c
11 October 2011	M/4643/2011	H4N6	10	10	8b	9B	4E	2c
21 October 2008	M/3661/2008	H4N6	12	13	7	12	6	3d
4 October 2011	M/4518/2011	H4N6	12	9B	5a	9B	5e	2c
4 October 2011	M/4528/2011	H4N6	12	4C	5a	10d	5e	4c
19 October 2011	M/4641/2011	H4N6	12	4C	5a	10d	5e	4c
16 November 2010	M/4182/2010	H5N3	11	4A	2a	10d	3	3d
19 November 2013	M/4971/2013	H5N3	4	4C	6b	10d	10	2b
26 November 2013	M/4952/2013	H5N3	11	4A	5b	10b	6	2b
1 September 2009	M/3720/2009	H6N2	6	4A	2b	3	12	B11
1 September 2010	M/4031/2010	H6N2	12	4A	2a	13	13	3c
11 October 2006	gull/M/3100/2006	H6N2	1c	10	7	1	6	6b
4 November 2008	M/3641/2008	H11N9	12	9A	3b	5	11	3d
21 October 2019	M/5712/2019	H11N6	11	12A	5a	10b	4E	3e

^a^ With the exception of one gull isolate (gull/M/3100/2006), all IAVs were isolated from ducks. The duck viruses are designated by isolation place (M), strain number, and isolation year, for example, M/5743/2019 stands for A/duck/Moscow/5743/2019. The viruses are ordered according to the HA subtype. ^b^ Clade numbers and subclade letters of the indicated segments. IAV isolates containing all gene segments of the same subclade are highlighted in yellow, with the strain names highlighted in orange.

**Table 2 viruses-13-01010-t002:** Diversity of influenza viruses of different subtypes ^a^.

	Duck Viruses Isolated in Moscow and The Netherlands							Gull Viruses	
	H1	H2	H3	H4	H5	H6	H11	H13	H16N3
NI	24	13	37	31	19	34	9	252	94
NG	20	11	34	29	18	29	9	84	38
%	83	85	92	94	95	85	100	33	40

^a^ NI, number of isolates; NG, number of genotypes; %, percentage of NG relative to NI.

**Table 3 viruses-13-01010-t003:** Genome constellations of IAVs isolated in The Netherlands from black-headed gulls in the years 2007–2016.

Strain Number	Clade/Subclade ^a^						Strain Number	Clade/Subclade					
	PB2	PB1	PA	NP	MP	NS		PB2	PB1	PA	NP	MP	NS
	bhg/N/xx/2007|H13N6 ^b^							bhg/N/xx/2007|H16N3					
2	13	5	8	14	11	16a	8	1	2b	8	3	3f	15
4	13	5	8	14	11	16a	3	1	2b	8	3	3f	15
6	13	5	8	14	11	16a	9	1	2b	8	3	3f	15
10	13	5	8	14	11	16a	7	13	5	8	4	1	15
	bhg/N/xx/2008|H13N8							bhg/N/xx/2008|H16N3					
15	2b	2a	9	6	2b	6	84	2a	1a	10b	13	3a	14a
55	2b	2a	9	6	2b	4	56	2a	1a	10b	13	3a	14a
51	3c	2a	11b	13	2b	6	33	1	1a	8	13	3a	14a
52	3c	2a	9	13	2b	6	92	2a	1a	10b	13	3b	14a
	bhg/N/xx/2009|H13N2, N6, N3							bhg/N/xx/2009|H16N3					
27 (N2)	11a	3d	10a	3	11	2c	35	11c	3d	10b	7	3b	2e
38 (N2)	3a	1b	10b	7	5	16b	25	11c	3d	10b	15	3b	2e
31 (N2)	12	6b	6a	2	10	16b	21	11c	3d	10b	15	3b	2e
39 (N6)	12	6b	6a	2	10	16b	28	11c	3d	10b	15	3b	2e
17 (N3)	11a	6b	11c	11	10	16b	23	11a	3a	11c	11	3c	2c
	bhg/N/xx/2011|H13N8, N3						22	11c	3b	10c	15	3b	2e
19	11c	3c	10c	15	7c	11		bhg/N/xx/2011|H16N3					
9	11c	3c	10c	15	7c	11	28	11c	3c	10c	9	3b	2e
39	11c	3c	10c	15	3d	8		bhg/N/xx/2011|H16N3					
8 (N3)	11b	10	10a	15	3d	2c	33	11b	10	10a	15	3d	2c
	bhg/N/xx/2012|H13N6						30	11b	10	10a	15	3d	2c
96	8a	8e	3b	8d	8e	9a	34	11b	10	10a	15	3d	2c
97	8a	8e	3b	8d	8e	9a	27	11b	10	10a	15	3d	2c
	bhg/N/xx/2014|H13N6						1	11b	10	10c	15	3d	2c
11	8c	9b	6b	8c	3c	10d	31	11b	10c	10a	15	3d	2c
15	8c	9b	6b	8c	3c	10d		bhg/N/xx/2012|H16N3					
28	8c	9b	6b	8c	3c	10d	107	10	3c	6d	15	3c	2c
3	8c	9b	6b	8c	3c	10d	114	10	3c	6d	15	3c	2c
32	8c	9b	6b	8c	3c	10d		bhg/N/xx/2013|H16N3					
16	10	9b	6b	8c	3c	10d	5	7c	8a	3c	8b	3d	9a
	bhg/N/xx/2014|H13N2						2	7c	8a	3c	8b	3d	9a
2	5	9a	5a	8c	6	9a	3	7c	8a	3c	8b	3d	9a
6	5	9a	5a	8c	6	9a	4	7c	8a	3c	8b	3d	12
37	5	9a	4	8a	6	16b		bhg/N/xx/2015|H16N3					
24	5	9a	4	8a	6	16b	1	7c	9b	5c	8c	3d	9a
	bhg/N/xx/2016|H13N2							bhg/N/xx/2016|H16N3					
11	9	8a	5b	8c	7b	16b	1	5	12	3b	8a	3d	9a
2	9	8a	5b	8c	7b	16b	3	5	12	3b	8a	3d	9a

^a^ Clade numbers and subclade letters of the indicated segments. IAV isolates containing all gene segments with matching subclades are depicted by orange color. Genome subclades highlighted in yellow are present in both H13 and H16 IAVs. Green color highlights non-persistent reassortment events. ^b^ Group name of the IAVs grouped in accord with isolation year and gene constellation. bhg, black headed gull; N, Netherlands; xx, strain number (specific strain numbers are listed in the first column).

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
