# Peer review of "Diversity and Reassortment Rate of Influenza A Viruses in Wild Ducks and Gulls"

_viruses, 2021, doi:10.3390/v13061010_

Round 1

Reviewer 1 Report

The article is interesting before; it could be considered for further publication. I have some queries that authors need to incorporate and revise their Manuscript.

Sentence formation needs crosscheck. Grammatical mistakes need to be minimized.

Major problem is that the study has been performed on samples collected from one city.

Line 38-41. Revise these sentences or better combine the two sentences.

Line 41 -44: Revise these sentences.

Line 45-50: The entire paragraph is missing supporting references.

Line 68: Mention the two lineages in a sentence.

Line 73-75: Revise this sentence.

Line 77-79: Confusing, grammatical mistakes, and confusion. Kindly revise.

Line 80 to 84; The entire paragraph needs to be revised. Avoid small sentences.

Line 108: Maybe you have used the same method of Seo et al. You can cite the supporting reference https://doi.org/10.1111/j.1365-2672.2012.05326.x

Line 329 to 333: Conclusion is concise; kindly write it in one paragraph.

Overall, the manuscript needs editing and proofreading, besides the citations of supporting references.

Author Response

Point 1:  Sentence formation needs crosscheck. Grammatical mistakes need to be minimized.

Response 1: The manuscript was additionally edited by a person with a good command of English.

Point 2: Major problem is that the study has been performed on samples collected from one city.

Response 2:

We agree with the reviewer. Therefore, in addition to viruses isolated in Moscow, we included about 500 viruses isolated in the Netherlands.  No changes in the manuscript were made in response to this comment.

Point 3: Line 38-41. Revise these sentences or better combine the two sentences

Response 3:

 We combined the sentences as follows 

IAVs of H1- H12, H14 and H15 HA subtypes mainly circulate in ducks; these IAVs replicate in the intestine, cause  asymptomatic infections, and transmit by the fecal-oral route through the water [1].                (Line 36-37)

Point 4:  Line 41 -44: Revise these sentences.

Response 4: We revised these sentences as follows:

IAVs of H13 and H16 HA subtypes mainly circulate in gulls, causing epidemics among chicks in densely populated breeding colonies. These gull IAVs replicate in the intestine and the respiratory tract, and they can transmit both by the fecal-oral route and through the air [3, 4, 5]. Thus, the patterns of replication and transmission differ between IAVs of ducks and gulls.           (Line 38-42)

Point 5: Line 45-50: The entire paragraph is missing supporting references.

Response5: We added references as shown below.

IAVs in domestic poultry evolve from IAVs of wild aquatic birds. IAVs of poultry can occasionally transmit and initiate host-specific evolutionary lineages in pigs, humans and other mammals [1, 2]. Long-term adaptation in ducks enables efficient replication and release of the virus into the environment without significant pathogenic effects to the host [2]. In gulls, the virus also replicate asymptomatically, while in poultry it may evolves towards increased pathogenicity [3]. In mammals, the influenza virus usually causes non-fatal disease [1].                                (Line 43-48)

Point 6: Line 68: Mention the two lineages in a sentence.

Response 6: We modified the sentence as follows.

In horses two lineages of influenza viruses (H7N7 and H3N8) evolved for decades [27].

           (Line 66)

Point 7: Line 73-75: Revise this sentence.

Response7: We modified the sentence as follows.

Introduction of the H5N1 IAVs into ducks and their reassortment with duck viruses of other subtypes led to the emergence of novel H5N3, H5N6 and H5N8 IAV lineages [31-33].

           (Line 71 - 73)

Point 8: Line 77-79: Confusing, grammatical mistakes, and confusion. Kindly revise.

Response 8: We revised this paragraph as follows.

It is impossible to detect persistent evolutionary lineages of duck IAVs  containing stable constellations of all eight  gene segments as duck IAVs constantly exchange segments during mixed infections with neither significant fitness costs nor selective advantages of newly emerging combinations of segments [34].                      (Line 75-77)

Point 9:  Line 80 to 84; The entire paragraph needs to be revised. Avoid small sentences.

Response 9:  This paragraph was revised as follows.

IAVs of almost all subtypes were found in gulls. However, the H1-H12 and H14 subtypes were only rarely isolated from gulls, and , as a rule, the gull viruses were evolutionarily close to the contemporary duck IAVs that circulated in the same region.  These cases can be classified as occasional spillovers of duck H1-H12 and H14 IAVs to gulls. By contrast, the IAVs of H13 and H16 subtypes were almost exclusively detected in gulls and terns [35-37].                     (Line 78-82)

Point 10: Line 108: Maybe you have used the same method of Seo et al. You can cite the supporting reference https://doi.org/10.1111/j.1365-2672.2012.05326.x

Response 10:

We would prefer not to add this reference to avoid renumbering of the literature. We think that the text describes a routine procedure and is sufficient without references.  

Point11: Line 329 to 333: Conclusion is concise; kindly write it in one paragraph.

Response 11: Done.                  (Line 333)

Point 12: Overall, the manuscript needs editing and proofreading, besides the citations of supporting references.

Response 12: All mentioned errors and obscure  sentences are corrected, four additional references are included in the text. The manuscript was additionally edited by a person with a good command of English.

Reviewer 2 Report

In the manuscript by Postnikova et al. the authors investigated the genetic diversity and reassortment rate of influenza A viruses in wild ducks and gulls. The authors analysed the complete genomes of 38 viruses isolated from feces of ducks and gulls of a pond in Moscow city between 2006-2019. Further, for comparison the authors analysed sequences from influenza A viruses isolated from wild ducks and gulls in the Netherlands in the same years. The study demonstrates that the diversity of influenza viruses is very high in ducks and no stable genotypes persist. In contrast, the diversity of viruses was much lower in gulls. The authors discuss that among other factors the density of colonies during breeding may account for the observed differences in genetic diversity of influenza viruses in ducks and gulls.  

The study is very interesting and extends the current knowledge on the genetic diversity and evolution of influenza viruses in birds. The manuscript is well written and the data are clear presented and discussed.

Few minor points that may be addressed by the author:

Do individual duck species (e.g. mallard duck versus shoveler versus teal) vary in their genetic diversity of influenza A viruses? Which duck species were predominant in this study?

Is it possible that subtypes H13 and H16 interfere with co-infection of gulls with other HA subtypes and thereby contribute to the significantly reduced genetic diversity and reassortment rate in gulls?

Do ducks and gulls vary in migration and could this contribute to the differences observed in genetic diversity in ducks versus gulls?

Figure 1: The font size of the clades/subclades should be increased.

Page 13: Lines 324-326. This sections was probably pasted by mistake to the discussion section.

Author Response

Point 1: Do individual duck species (e.g. mallard duck versus shoveler versus teal) vary in their genetic diversity of influenza A viruses? Which duck species were predominant in this study?

Response 1:

All viruses studied in our work were isolated from one species of ducks (mallard) and one species of gulls (black-headed gull). Corresponding explanations are included in the text (Line 101, 157, 176)

Point 2: Is it possible that subtypes H13 and H16 interfere with co-infection of gulls with other HA subtypes and thereby contribute to the significantly reduced genetic diversity and reassortment rate in gulls?

Response 2:

We agree with the reviewer that infection of gulls with viruses of the H13 and H16 subtypes may prevent co-infection with viruses of other HA subtypes. Furthermore, as discussed in the text a poor fitness of duck-origin gene segments in gulls may reduce viability of reassortant viruses containing these segments and thus represents another reason for low genetic diversity of gull viruses 

(line299-327).

Point 3: Do ducks and gulls vary in migration and could this contribute to the differences observed in genetic diversity in ducks versus gulls?

Response 3:

Yes, the way ducks migrate facilitates reassortment of influenza viruses, while gulls do not form large flocks during migration. Relevant explanations are included in the text

 (line 288).

Point 4: Figure 1: The font size of the clades/subclades should be increased.

Response 4: The exact designations of the clades/subclades are given in Tables 1 and 3, as well as in Tables S2, S3 and S4 of supplemental materials. We would prefer not to add this information to the Figure 1, as this will make it too busy.

Point 5: Page 13: Lines 324-326. This sections was probably pasted by mistake to the discussion section.

Response 5: We are very grateful to reviewer for noticing this error.  We deleted the text in the lines 324-326.

Round 2

Reviewer 1 Report

The authors have addressed my comments.